# Neurodegeneration with Brain Iron Accumulation Disorders: Valuable Models Aimed at Understanding the Pathogenesis of Iron Deposition

**DOI:** 10.3390/ph12010027

**Published:** 2019-02-09

**Authors:** Sonia Levi, Valeria Tiranti

**Affiliations:** 1School of Medicine, University Vita-Salute San Raffaele, 20132 Milano, Italy; 2San Raffaele Scientific Institute, Division of Neuroscience, 20132 Milano, Italy; 3Fondazione IRCCS Istituto Neurologico Carlo Besta, 20126 Milano, Italy; Valeria.Tiranti@istituto-besta.it

**Keywords:** iron, neurodegeneration, NBIA

## Abstract

Neurodegeneration with brain iron accumulation (NBIA) is a set of neurodegenerative disorders, which includes very rare monogenetic diseases. They are heterogeneous in regard to the onset and the clinical symptoms, while the have in common a specific brain iron deposition in the region of the basal ganglia that can be visualized by radiological and histopathological examinations. Nowadays, 15 genes have been identified as causative for NBIA, of which only two code for iron-proteins, while all the other causative genes codify for proteins not involved in iron management. Thus, how iron participates to the pathogenetic mechanism of most NBIA remains unclear, essentially for the lack of experimental models that fully recapitulate the human phenotype. In this review we reported the recent data on new models of these disorders aimed at highlight the still scarce knowledge of the pathogenesis of iron deposition.

## 1. Introduction

The acronym NBIA (neurodegeneration with brain iron accumulation) designates a clinically and genetically heterogeneous group of neurodegenerative diseases, the majority of which is hallmarked by iron deposits in the brain [1]. Iron accumulation, is detected in vivo by MRI analysis, and affects mainly the basal ganglia regions, such as globus pallidus (*GP*) and substantia nigra (*SN*), although other regions including the cortex and cerebellum can be involved. These disorders are inherited as Mendelian traits (autosomal recessive, dominant, or X-linked), and typical features are extra pyramidal symptoms, namely, dystonia, parkinsonism, spasticity, variably associated with neuropsychiatric abnormalities, and optic atrophy or retinal degeneration [1]. Disorders belonging to the NBIA group are rare, between one and three per million individuals in the general population, and 15 causative genes, whose products are involved in a wide spectrum of biological activities, have been so far discovered. Although the presence of brain iron accumulation enables the diseases to be included in the NBIA group, the pathogenic mechanisms linking specific disease-genes mutations to iron metabolism are unclear [2]. *CP* (ceruloplasmin) and *FTL1* (ferritin light chain) are directly associated to iron homeostasis (Table 1). The remaining genes (*PANK2*, *COASY*, *PLA2G6*, *C19orf12*, *FA2H*, *ATP13A2*, *WDR45*, *DCAF17*, *SCP2*, *GTPBP2*, *AP4M1*, *REPS1*, and *CRAT*) are unrelated to iron metabolism, being involved in diverse metabolic pathways (Table 1). 

*PANK2* (pantothenate kinase type 2) and COASY (CoA synthase) code for enzymes in the crucial biochemical pathway responsible for coenzyme A (CoA) production. *PLA2G6* (phospholipase A2 group 6), *FA2H* (fatty acid hydroxylase 2), *C19orf12*, *SCP2* (sterol carrier protein type 2) and *CRAT* (carnitine acetyltransferase) genes code for proteins involved in lipid metabolism, membrane integrity and mitochondrial function. *WDR45*, *ATP13A2*, and possibly *AP4M1* are genes coding for factors involved in the autophagic process. The *GTPBP2* gene product plays a putative role in chain elongation during protein synthesis at the ribosome, *REPS1* (RALBP1-associated EPS domain-containing protein 1) is involved in endocytosis and vesicle transport, while the *DCAF17* (also known as *C2orf37*) gene encodes for a protein with still not clarified cellular localization, probably in the nucleolus, and with undefined activity (Figure 1). Around one third of the NBIA causative genes namely *PANK2*, *COASY*, *PLA2G6*, *C19orf12*, and *CRAT*) are associated to mitochondria, suggesting that these organelles play a crucial role in triggering the disorders. Thus, these pathologies represent an unprecedented chance to investigate the molecular mechanisms involved in iron management in the brain and the connection with mitochondrial function and neurodegeneration. In this review, we will focus on what we have learned from available in vivo and in vitro models about iron metabolism and its connection with the different underlying genetic mutations of NBIA. 

### 1.1. Iron Homeostasis in Diseases Caused by Mutations in Iron-Related Genes 

#### 1.1.1. *CP* (OMIM*117700)

Aceruloplasminemia (OMIM #604290) is the first disease in which the alteration of an iron-related protein has been linked to neurodegeneration. It is inherited as an autosomal recessive trait and was originally described in a female from Japan [3] where the prevalence of mutations is around one in 2,000,000 individuals. The genetic analysis of patients affected by aceruloplasminemia revealed about 50 distinct causative mutations (listed in http://www.hgmd.cf.ac.uk/ac/index.php) scattered along the gene and identified in more than 60 families [4,5]. *CP* gene is located on chromosome 3q21-24 and encodes ceruloplasmin (Cp), a plasma membrane glycoprotein with ferro-oxidase function containing many copper atoms, which facilitates ferroportin-mediated cellular iron export [6]. 

Onset of symptoms is in adulthood and the clinical features include neurological signs like blepharospasm, grimacing, facial and neck dystonia, tremors, chorea and ataxia, and cognitive dysfunction. Additional signs, generally preceding neurological symptoms are diabetes mellitus, retinal degeneration and microcytic anemia [5]. The disease is also hallmarked by alteration of hematological parameters such as: very low or undetectable serum ceruloplasmin, high levels of serum ferritin and low levels of iron [7]. Iron depositions affect the striatum, thalamus, and dentate nucleus as showed by abnormal low intensities on T1 and T2 weighted images obtained by MRI analysis [8]. Neuropathology investigations demonstrate aberrant basal ganglia and dentate nucleus, with consistent iron accumulation in neuronal and glial cells [9,10,11,12]. Characteristic neuropathological findings are enlarged or deformed astrocytes and spheroid-like globular structures caused by iron accumulation localized to terminal astrocytic processes [11]. Iron overload was also detected in the cerebral cortex, retina, and cerebellum, and in non-cerebral tissues, such as the pancreas and myocardium [12]. 

Mutations of *CP* determine structural modifications of the protein (Cp) which are responsible for retention of Cp in the ER, miss-incorporation of copper atoms into apo-ceruloplasmin or defective ferro-oxidase activity [13,14,15]. Overall, this malfunctioning causes impairment of iron export from the cell, which in the end is responsible for iron overload. 

Investigations of the pathogenetic mechanisms underlining aceruloplasminemia were performed by analyzing the effect of Cp mutants’ expression in mammalian cell culture [13,14,15] and by characterizing various aceruloplasminaemia mouse models [16,17,18]. Overall these data indicated that, in the presence of an aberrant ceruloplasmin, iron entering the CNS as ferrous iron could not be oxidized. As a consequence, cells can internalize large quantities of ferrous iron through non-regulated internalization pathways of non-transferrin-bound iron [19]. Accumulation of iron inside astrocytes observed in the pathology is due not only to the excess of import, but also to the inability of the cells to export iron though ferroportin, in the absence of ceruloplasmin. In this way iron would remain “locked” in the astrocytes and could not reach the neurons, which would paradoxically die from iron deficiency and from toxic compounds released by degenerating astrocytes. Additionally, it has been proposed that some mutated Cp forms could also accumulate in aggregates leading to death of astrocytes through an iron-independent pathway [20]. 

The specific degeneration of astrocytes and neurons is explained by their strict dependence on ceruloplasmin ferro-oxidase activity. On the contrary, other cells in the CNS, including oligodendrocytes, depend on the action of hephaestin, a protein similar in structure and function to ceruloplasmin. In fact, only the simultaneous ablation of *CP* and hephaestin (*HEPH*) genes in mice is able to recapitulate symptoms consistent with those shown by aceruloplasminemia patients, suggesting that hephaestin can compensate the lack of CP in mice, but not in humans [21,22]. This double knock-out mouse model showed iron accumulation in gray and white matter oligodendrocytes [22], macular degeneration, iron overload, and increased oxidative stress in the retina [23]. Treatment of these mice orally with DFP showed to be protective against oxidative stress, retinal degeneration and accumulation of lipofuscin, and was able to reduce ataxia and increase life span [24]. Despite the good results obtained in mice, iron chelation therapy in patients was disappointing since it only partially reduced systemic iron deposition in liver and pancreas [25], but it was ineffective in reducing the brain iron accumulation and thus in controlling neurodegeneration [12,26]. 

Another proposed therapeutic approach is based on enzyme replacement therapy (ERT) with ceruloplasmin. Recently, it was demonstrated that the parental administration of human ceruloplasmin to *CP*-/- mice was able to restore normal levels of the protein in the brain, as well as its ferro-oxidase activity [27]. Interestingly, in treated mice, the reduced brain iron deposition led to motor coordination improvement, suggesting ERT as a possible therapeutic option for aceruloplasminemia [27]. 

#### 1.1.2. *FTL* (OMIM*134790)

Neuropherritinopathy (NF) is an extremely rare autosomal dominant disease described for the first time in 2001 [28] and linked to mutations of *FTL1*, the gene coding for L-ferritin subunit that, together with the H-ferritin subunit, forms the major iron storage protein. 

The most frequent mutation is the 460InsA (or c.460dupA) described in 41 cases [29] presenting with focal onset chorea or focal dystonia and history of movement disorders. Serum ferritin level was generally low. MRI was abnormal in all reported cases and showed lesions in the globus pallidus, putamen, and dentate nuclei. Few cases of NF presented with the “eye of the tiger” sign (typical of PKAN), a hypointense MRI signal, which is pathognomonic for iron accumulation, surrounding a central region of hyperintensity, which is due to gliosis and/or necrosis, in the anteriomedial globus pallidus on the T2* sequence [8]. A typical pathological feature found in astrocytes and oligodendroglia in the grey and white matter, was the presence of intranuclear and intracytoplasmic bodies containing ferritin and iron in insoluble form. The same bodies were also present in fibroblasts, tubular epithelium of the kidney, and endothelial cells of muscle capillaries. Neurons and glia also showed swollen or vacuolated nuclei containing iron and ferritin [30]. 

Biochemical studies performed by overexpression of mutants 460InsA and 498InsTC (or c.497_498dupTC) in human cells, demonstrated instability of mutant proteins, a different solubility and a lower capacity to incorporate iron as compared to wild-type ferritin [31,32,33,34,35]. These cellular models showed that the expression of these ferritin mutants not only increased cellular free iron pool, but also induced oxidative damage and the impairment of proteasome activity [33,34].

Brain iron accumulation and increased production of reactive oxygen species, are present in mouse models generated by transgene expression of *FTL1* cDNAs coding for variants of the human ferritin protein. These mice showed features of neurological deterioration, decreased motor activity and coordination, reduced lifespan together with the presence of ferritin aggregates in the nucleus and cytoplasm [36,37,38,39]. 

Specifically, mice overexpressing the human mutant c.497_498dupTC recapitulate features typical of the human disease, including abnormalities of brain iron homeostasis [36,37]. Further investigation of post-natal hippocampal neurons in transgenic mice with a different genetic background, revealed high susceptibility to chronic iron overload and/or acute oxidative stress in comparison to wild-type neurons [37]. Moreover, ultrastructural analyses of specific brain regions such as cerebellum and striatum, showed an accumulation of lipofuscin granules associated with iron deposits, particularly evident in aged mice [37]. Altogether, these data indicate that the combination of increased oxidative stress and lipofuscin accumulation play a key role in the etiopathogenesis of human NF. The only data available on human naïve fibroblasts derived from a patient carrying the c.497_498dupTC, demonstrated a consistent increase of intracellular iron, altered iron management, ferritin accumulation and the presence of oxidative stress markers, as previously demonstrated in patients and transgenic mice [40].

From the overall data, the emerging hypothesis suggests that the decreased ability of ferritin to sequester iron excess has two major consequences: (i) the unbound metal precipitating on proteins promotes their aggregation [31]; (ii) the higher free iron-dependent oxidative damage required the need for the cell to increase degradation of oxidized molecules. Both these effects cause the engulfing of the cellular recycling systems (i.e., proteasome and lysosome), and in the long-term leads to cellular death. These events occur despite the regulation system of iron entry into the cell seeming to work properly [33]; in fact, the level of the main iron-importer protein, the transferrin receptor, appears strongly reduced in different experimental models [32,33,34]. Thus, apparently, iron may be efficiently internalized by transferrin/transferrin-receptor independent pathway in neuronal compartment. This pathogenetic mechanism seems to be harmful mainly in non-dividing cells, partly explaining the specificity of the symptomatology. Interestingly, since the rate of lipofuscin accumulation is known to correlate negatively with longevity, the NF pathogenesis may replicate the cascade of events that occur during the physiological process of aging.

The most simple and rational way to reduce iron accumulation is the utilization of iron-chelating compounds such as desferrioxamine (DFO) or deferiprone (DFP). These treatments were tested both in animal models and in patients, with comparable results. DFP treatment in mice was effective in reducing iron accumulation in the brain [41] and had significant effects on systemic iron homeostasis [42]. As already discussed for the aceruloplasminemia, also in this case the outcome of the treatment with iron chelators is not as effective in humans. Iron-chelating based therapies in NF patients had severe detrimental effects on systemic iron level causing iron depletion without improving clinical conditions [29,43]. This may be due to a lower penetrability of the iron chelator in the basal ganglia of human and/or to a late treatment in humans, which develop symptoms when the brain region has already degenerated.

### 1.2. Iron Homeostasis in Inborn Errors of Coenzyme A Biosynthesis

#### *PANK2* (OMIM*606157) and *COASY* (OMIM*609855)

Mutations in genes coding for the first (*PANK2*) and the last (*COASY*) enzyme of the biosynthetic pathway that produces Coenzyme-A (CoA) cause NBIA subtypes known as pantothenate kinase-associated neurodegeneration (PKAN) and CoA synthase protein-associated neurodegeneration (CoPAN), respectively. 

PKAN accounts for a large fraction of NBIA cases [44] while CoPAN appears more rare, being so far identified in few individuals worldwide [45,46,47,48]. These two disorders display an impressively similar phenotype, presenting with early-onset spastic-dystonic paraparesis with a later appearance of parkinsonian features, cognitive impairment, obsessive-compulsive disorder, and brain iron accumulation [45]. PKAN typically manifests in early childhood with gait disturbances and rapidly progresses to a severe movement deficit with dystonia, dysarthria and dysphagia [49]. The hallmark of PKAN is the “eye-of-the-tiger” signal in the globus pallidus on T2-weighted magnetic resonance imaging, which reflects the focal accumulation of iron in this area [50]. CoPAN also accumulates iron in *GP*. The *SN* is as well affected in CoPAN and to a lesser extent in PKAN. The iron accumulation correlates with neural damage. It is often associated with axonal expansions (spheroids), but it also appears as a granular form in perivascular location [51]. Although iron accumulation hallmarks PKAN and CoPAN, its relationship with CoA dysfunctional biosynthesis is not clear. 

Despite several efforts have been made during the last years to understand the connection between iron homeostasis and mutations in *PANK2* or *COASY*, no models, either in vitro or in vivo, have been so far able to recapitulate the presence of iron deposits. *Pank2* null mice showed growth reduction, azoospermia [52] and impaired mitochondrial function [53], but did not show signs of neurodegeneration or iron accumulation and did not display movement disorders. Only by increasing the fat content in the diet of these mice it was possible to unravel a neurodegenerative phenotype characterized by motor dysfunction, bioenergetic failure, and brain cytoplasmic accumulation of abnormal ubiquinated proteins [54], features observed in the brains of PKAN patients [51]. The simultaneous knockout of *Pank1/Pank2* led to severe reduction of CoA levels in the brain but iron accumulation was not reported [55]. The combined ablation of *Pank2/Pank3* or *Pank1/Pank3* in mice is associated with a lethal phenotype [55]. Altogether, these experiments imply that, in mice, *Pank2* loss is counterbalanced by the other *Pank* genes, but the simultaneous elimination of *Pank3* with either *Pank2* or *Pank1* is incompatible with life. 

*Drosophila* possesses only one pantothenate kinase isoform (*fumble*) and its ablation determines brain vacuolization with defective neurological functions but without brain iron overload [56], resulting in flies with severe motor impairment [57]. The down-regulation of *pank2* in Zebrafish induced defects in neuronal development and an aberrant organization of the vascular system [58]. 

PKAN patients’ fibroblasts have been helpful to reveal some defects in mitochondrial activity and iron metabolism associated with *PANK2* deficiency [59], however, the pathological neurodegenerative processes cannot be appreciated in these cells. 

Insight on the neurodegenerative process started emerging thanks to the generation of iPSC-derived glutamatergic neurons from PKAN patients and from healthy subjects. Their compared analysis showed alteration of mitochondria functionality in PKAN patients. The altered phenotype included impairment of mitochondrial iron-dependent biosynthesis that caused iron dys-homeostasis and consequent enhanced ROS production, leading to major membrane excitability defects. In particular, the cells inability to maintain Fe/S biosynthesis simulated a cellular iron deficient phenotype that promoted cellular iron uptake increasing the translation of the TfR1 [60]. 

A second paper reported data on iPSCs derived cortical neuronal cells obtained from fibroblasts of an atypical PKAN patient [61], in which the PANK2 protein was not completely absent but reduced as compared to control cells. In this case, CoA homeostasis and cellular iron handling were normal, but mitochondrial functionality resulted affected, with increased production of reactive oxygen species and lipid peroxidation [61]. 

Neither glutamatergic nor cortical neurons displayed iron deposits and iron chelation in cortical neurons exacerbated the mitochondrial phenotype in both control and patient neuronal cells [61]. Thus, both these investigations on human neurons agree to conclude that mitochondrial impairment is an early feature of the disease process and the consequent ROS formation a main actor in the pathogenesis, while iron chelation treatment did not seem a good option. Data on efficacy and safety of deferiprone for the treatment of PKAN were available through clinical trials on small cohorts of patients and showed controversial results [50,62].

Data on the efficacy of DFP treatment collected in a wide clinical trial involving around 100 PKAN patients in the world are still under analysis and will be available shortly (https://tircon.eu). While DFP efficacy is still under investigation, the work performed in vitro on glutamatergic neurons demonstrated that CoA supplementation prevented neuronal death and reduced ROS formation by restoring mitochondrial and neuronal functionality, suggesting CoA treatment as a possible therapeutic intervention [60]. 

Very recently, an allosteric PANK activator has been identified by compounds library screening. The compound denominated PZ-2891 crosses the blood brain barrier and oral administration to wild-type mice increases coenzyme A levels in the brain and liver. Moreover, PZ-2891 treatment of a knockout mouse model of brain CoA deficiency presenting with weight loss, severe locomotor impairment and early death, was able to rescue the weight, to improve locomotor activity and to extend life span [63].

Analysis of fibroblasts derived from CoPAN patients, showed impaired, but not completely abolished, de-novo synthesis of CoA and dephospho-CoA. This would suggest the presence of alternative routes for CoA production, or the preservation of a residual catalytic activity of the mutant CoA synthase protein [45]. More recently, mutations of *COASY* associated with the complete absence of the protein were reported in two cases of pontocerebellar hypoplasia, microcephaly, and arthrogryposis [48] with an invariable lethal phenotype, probably due to the complete absence of CoA.

In the yeast *Saccaromices cerevisiae*, CoA synthesis is carried out by five sequential enzymes (CAB1 to CAB5) deletion of which, leads to a lethal phenotype. Re-expression of wild-type yeast CAB5 or human COASY protein was able to rescue the lethal phenotype while mutant CAB5 or COASY expression determined a phenotype characterized by reduced growth, auxotrophy for panthotenate, and decreased amount of CoA in isolated mitochondria [45]. Additional features included mitochondrial dysfunctions characterized by impairment of the respiratory chain and revealed iron accumulation and impaired lipid content [64]. 

In zebrafish the complete down-regulation of Coasy, obtained by morpholino-mediated approach, was associated with impaired development and premature death, while a partial down-regulation gave rise to a milder phenotype [65]. Main phenotypic features of morphants included: reduced size of the brain with less defined brain structures, disorganization of the vascular system in the brain and trunk, and reduced levels of CoA in the fish embryos. Rescue of these features was obtained by adding CoA to the fish water or by re-expression of the human wild-type gene, indicating a causative role of impaired CoA synthesis in determining the pathological phenotypes. 

Hence, the connection between COASY dysfunction and iron metabolism needs to be clarified although a potential link was highlighted by the identification of an iron responsive element (IRE) in the 3’ region of this gene that could stabilize the COASY mRNA in the presence of iron [2].

Mutations in *PPCS* encoding for the second enzyme in the CoA biosynthetic pathway converting 4’-Phosphopantothenate into phosphopantothenoylcysteine, have been described in two unrelated families showing a very severe pediatric phenotype characterized by dilated cardiomyopathy associated with reduced level of CoA [66]. In the affected individuals neurodegeneration and iron accumulation were not investigated or not present because of a premature death of some patients. At the moment, it is not clear why *PPCS* mutations lead to cardiac alteration instead of neurodegeneration and further investigation will be required. We have decided to mention this gene for the sake of completeness despite that *PPCS* cannot be included in the NBIA disease genes, at least from the knowledge available at the time of writing.

### 1.3. Iron Homeostasis and Mutations in Lipid Metabolism Related Genes

Globally these sub-types of NBIA disorders typically begin in childhood with a combination of clinical signs, which are also present in spastic paraplegia complex forms and in several cases the genetic cause of the disease is shared by the two category of diseases. The accumulation of iron is not a consistent finding and is often documented in advanced disease stages suggesting that it could contribute to disease progression but is not the primary cause [67].

#### 1.3.1. *PLA2G6* (OMIM*603604)

*PLA2G6*-associated neurodegeneration (PLAN) is an autosomal recessive heterogeneous group of related neurodegenerative conditions comprising infantile neuroaxonal dystrophy (INAD, OMIM #256600) and atypical neuroaxonal dystrophy (ANAD, OMIM #610217) caused by mutations in *PLA2G6*, mapping on human chromosome 22q13.1 and encoding for calcium-independent phospholipase A2 group VI (iPLA2β or iPLA2VI) [68]. 

Onset of disease is typically in childhood and presents as neuroaxonal dystrophy with psychomotor regression and axial hypotonia, accompanied by neuropathy with spheroidal bodies in the peripheral nerves, detectable by analysis of sural nerve biopsy. Cerebellar atrophy is detected in the majority of the cases by brain MRI while iron accumulation is present in only half of the affected individuals. PLA2G6 protein is ubiquitously expressed and is localized in the cytosol and mitochondria, playing a role in phospholipids metabolism and membrane remodelling. Malfunctioning of the protein may lead to alteration of neuronal membranes integrity and fluidity [69] determining neuronal suffering.

A *Pla2g6* knock-out mouse model recapitulate in a precise way the human disease and showed cerebellar atrophy, loss of Purkinje cells, pronounced neuroinflammation, age-dependent accumulation of spheroids but not iron accumulation, which is not, however, a consistent finding neither in the affected patients [69]. 

This mouse model was also characterized by the presence of ultrastructural alteration of mitochondria in the brain with a consistent accumulation of cardiolipin, a phospholipid formed from two phosphatidic groups connected by a glycerol molecule [70], which is highly enriched in the mitochondrial inner membrane. In fact, first pathological signs in neurons of *Pla2g6*-null mice is the disorganization of the cristae in the mitochondrial inner membrane with the consequent destruction of the mitochondrion, the release of cytochrome c, and the formation of swollen and degenerated axons as the disease progresses. Presynaptic membranes are altered in this animal model, demonstrating the role of PLA2G6 may be to remodel cardiolipin and eliminate fatty acids damaged by oxidative stress [71]. Recently, an age-dependent iron accumulation in *SN*, *ST*, and *GP* brain regions was reported on *Pla2g6* -/- mice. These mice showed an increased level of iron-proteins (TfR1, DMT1 and IRPs), decreased FPN1, enhanced lipid peroxidation and mitochondrial dysfunction. These alterations precede the sign of neuronal death, suggesting that iron-dependent ROS formation might be a primary cause of neurodegeneration [71].

The similarity between PKAN and PLAN phenotypes can be traced back to the fact that PANK2 and PLA2G6 enzymes would take part to cardiolipin synthesis or remodelling in the inner mitochondrial membrane [72].

#### 1.3.2. *FA2H* (OMIM*611026)

Fatty acid hydroxylase-associated neurodegeneration (FAHN) (OMIM #612319) was first described in two consanguineous families, of Italian and Albanian origin respectively and is a sub-type of neurodegeneration with brain iron accumulation transmitted as an autosomal recessive trait [73]. The gene responsible for the disorder is the fatty acid hydroxylase-2 (*FA2H)*, that maps on human chromosome 16q23.1 and codes for a NADPH-dependent mono-oxygenase localized on ER membranes [73,74]. The same gene is also mutated in cases of leukodystrophy and a form of spastic paraplegia [75]. Main clinical features of the disease are spasticity, ataxia, dystonia, optic atrophy, and oculomotor abnormalities followed by cognitive disorders and epilepsy. Magnetic resonance studies in patients have shown iron overload in the globus pallidus, substantia nigra, and subcortical and periventricular regions [76], as well as abnormalities of the white matter distribution, cerebellar atrophy, and thinning of the corpus callosum.

FA2H enzymatic activity is responsible for the hydroxylation of fatty acids, which are incorporated in the sphingolipids and its impairment causes myelination defects with the consequent impairment of the axonal function [77]. Mice lacking the *Fa2h* gene, either constitutively or limited to the glia cells, showed degeneration and loss of myelin sheath in the spinal cord and in sciatic nerves [77,78].

Finally, in the N-terminus of FA2H is present a cytochrome b5-like heme binding domain and a fatty acid hydroxylase domain with an active site for the binding of non-heme di-iron. The presence of these domains has been suggested to allow a direct interaction with iron-containing moieties, thus indicating a possible link between gene function and iron accumulation [73], although this hypothesis is not supported by experimental evidences. A recent in silico protein modelling report demonstrated that the FAHN causative missense mutations alter the heme-binding site or disrupt the hydroxylase domain impairing the enzyme catalytic activity. However, the mechanism by which reduced FA2H activity generates the iron deposition was not investigated [79].

#### 1.3.3. *C19orf12* (OMIM*614297)

Mutations in *C19orf12* cause a rare form of NBIA, transmitted as an autosomal recessive trait, denominated mitochondrial membrane protein-associated neurodegeneration (MPAN) (OMIM #624298). *C19orf12* maps on human chromosome 19 and encodes for a mitochondrial protein associated with the membranes and with a putative role in lipid metabolism [80,81].

Mutations in *C19orf12* cause other neurodegenerative disorders such as: pallido-pyramidal syndrome [82], hereditary spastic paraplegia phenotype 43 (SPG43) [83] and ALS [84]. The main clinical features of MPAN include: progressive spastic para/tetraparesis, dystonia, motor axonal neuropathy, parkinsonisms, psychiatric symptoms, retinal abnormalities, and optic atrophy [80,85,86]. Iron accumulation is present in the globus pallidus and substantia nigra, as highlighted by MRI analysis, showing T2-weighted hypo-intensity [87]. Neuropathological analysis of patients’ brains demonstrated the presence of axonal spheroids, Lewy bodies, axonal swellings, and hyperphosphorylated tau in the cortex, *GP*, caudate/putamen, *SN*, and brain stem. 

In silico evaluation of the *C19orf12* protein indicated that it contains a transmembrane glycine zipper. The protein has two isoforms deriving from different start codons: the longer isoform has multiple sub-cellular localization: it is present into mitochondria, endoplasmic reticulum [88], and a small fraction in the MAM, contact regions between mitochondria and ER functionally-relevant for phospholipids metabolism [88]. The protein has high levels of expression in the brain, blood cells and adipocytes, while transcriptome analysis indicated aco-regulation with genes involved in lipid metabolism with a possible connection with CoA synthesis [80]. The function of the protein is still not clarified and the simultaneous knock-down of two *C19orf12* orthologs CG3740 and CG11671 in *Drosophila*, demonstrated a defect in climbing activity, bang sensitivity, and the presence of vacuoles in the brain and optical lobe but the absence of iron accumulation [89]. 

#### 1.3.4. *SCP2* (OMIM*184755) and *CRAT* (OMIM*600184)

Mutations in the sterol carrier protein-2 (SCP2) gene (OMIM #613724) cause a complex phenotype characterized by leukencephalopathy, dystonia, torticollis, azoospermia, cerebellar ataxia, and gait impairment [90,91]. Brain MRI analysis showed bilateral hyperintense signals in the thalamus, indicating iron overload, butterfly-like lesions in the pons, and lesions in the occipital region. The *SCP2* gene is located on chromosome 1p32.3 and exons 1–16 encode for the sterol carrier protein X (SCPx), while exons 12–16 encode the sterol carrier protein 2 (SCP2) [92]. Both enzymes have thiolase activity required for the breakdown of branched chain fatty acids and are located into peroxisomes. The pathogenic effects are probably due to elevated levels of branched chain fatty acids in patients with SCP2 mutations [93]. Abnormal fatty-acid acyl-CoA metabolism is again suggested as the culprit for a disease belonging to the big group of NBIA, highlighting the role of lipids as a common pathogenic mechanism for this category of disorders. 

Mutations in *CRAT*, coding fora protein involved in the transfer of acyl groups from carnitine to coenzyme A and in the transport of fatty acids for beta-oxidation have been recently identified in one subject with an NBIA phenotype [94]. Intensive investigations of fibroblasts derived from patients with *CRAT* and *REPS1* [92] mutations as well as fibroblasts derived from patients with *PANK2*, *PLA2G6*, *C19orf12*, and *FA2H* mutations, led to the identification of a common pathogenic mechanism shared by the different diseases. Iron overload was found to be due to abnormalities in transferrin receptor (TFR1) recycling and reduction of its palmitoylation [94]. This is the first time that a common pathogenic mechanism has been, not only hypothesized, but also experimentally demonstrated. Interestingly enough, it was also showed that the antimalarian drug artesunate, was able to increase TfR1 palmitoylation and decrease ferritin levels in fibroblasts of different NBIA subgroups [94].

This potential therapeutic treatment of NBIA should be seriously taken into consideration and could also be relevant for other more common neurodegenerative disorders, such as Parkinson’s or Alzheimer’s, in which iron overload is a common finding.

### 1.4. Iron Homeostasis and Autophagosome/Lysosome Regulation

Three genes, whose mutations cause neurodegeneration with brain iron accumulation related to malfunctioning of the autophagosome and or lysosome activity, will be described in this chapter and include: *WDR45*, *ATP13A2*, and possibly *AP4M1*.

#### 1.4.1. *WDR45* (OMIM*300526)

Mutations in the WD repeat domain 45 (*WDR45*) gene are responsible for β-propeller-associated neurodegeneration (BPAN) (OMIM #300894), a very peculiar disorder mapping on X chromosome, in which the deleterious mutations are invariable de novo. Patients of both gender display similar symptoms characterized since male patients were shown to have somatic or germline mosaicism [95]. Onset of the disease is in childhood but remains somehow silent until adulthood when neurological deterioration manifests as dystonia, parkinsonism, cognitive decline, and seizures. Iron overload in *GP* and *SN* was revealed by MRI that also showed, on T1 sequence, a dark central band surrounded by a halo of hyperintense signal in *SN* and cerebral peduncles.

*WDR45* belongs to the WD40 scaffold-protein family involved in protein-protein interactions. It is the human ortholog of the yeast *atg18*, which has been demonstrated to be crucial for the formation of the autophagosome. In fact, the phosphotidylinositide-3-phosphate binding motif present in atg18 determines its binding to the ER membrane, allowing the formation of protein complex [96]. A possible link between WDR45 autophagosome and mitochondria has been suggested by the evidence that proteins coded by *atg* genes are found transiently associated to the outer membranes of mitochondria [97].

Recently, it was reported that a brain-specific *Wdr45* knock-out mouse model showed learning and memory defect, axonal swelling, and impaired autophagic flux. The hippocampus and caudate nucleus revealed the presence of aggregates, which stained positive for ubiquitin, but the presence of high levels of iron was not observed [98]. Finally, Ingrassia et al. [99] demonstrated in fibroblasts of two BPAN individuals under conditions of nutrient deprivation, up-regulation of the divalent metal transporter 1 (DMT1), which is essential for uptake of iron in the cells, and the simultaneous down-regulation of the Transferrin receptor (TfR1) causing accumulation of ferrous iron within the cells. TfR1 down-regulation is in line with what observed in other NBIA derived fibroblasts [94] and could again represent a common pathogenic mechanism unifying several different forms of the disease. It is possible that the presence of altered autophagy process, as observed in fibroblasts carrying *WDR45* mutations, could affect the removal of proteins such as DMT1, determining iron overload [99].

#### 1.4.2. *ATP13A2* (OMIM*610513)

Kufor-Rakeb syndrome (KRS), an extremely rare autosomal recessive disorder characterized by early onset parkinsonism, pyramidal signs, dementia, and eye movements alteration, belongs to the NBIA group since some patients show iron accumulation in basal ganglia at MRI analysis [100]. Mutations in *ATP13A2*, located on chromosome 1p36.13 and encoding for a P-type ATPase, are causative of this disorders but have also been reported in different neurodegenerative diseases, such as juvenile onset parkinsonism and dementia (PARK9), and Ceroid-Lipofuscinosis [101]. This protein works mainly in the lysosome membranes as a cation pump, although modulation of its expression results in a variety of effects ranging from reduced intracellular calcium levels in cortical neurons to mitochondrial fragmentation [102]. Patients’ fibroblasts showed increased fragmentation of mitochondrial network, disrupted mitochondrial DNA integrity, increased ROS production, reduced ATP production [103]. *ATP13A2* deficiency leads to lysosomal dysfunctions, which could ultimately cause impaired degradation of substrates and damage of autophagosome clearance [104]. Alteration of autophagy, cell death, and accumulation of -synuclein occur after down-regulation of ATP13A2 in dopaminergic neurons and primary cortical neurons derived from mice [105]. 

Ablation of *Atp13a2* gene in mice causes lipofuscin and α-synuclein accumulation, gliosis, increased level of ubiquitinated proteins, endolysosomal abnormalities, and deficiency of sensory and motor coordination probably due to dopaminergic pathology [106,107].

The only experimental evidence, demonstrating a direct role of this P-type ATPase in iron intracellular handling, was reported by Rajagopalan et al. [108]. The authors identified *ATP13A2* as a gene target of HIF1 and, therefore, it is subjected to the iron-dependent regulation of the PHD2-HIF1 signaling pathway [108]. This study suggests that the perturbation of lysosomes and acidic endosomes functions, which are important for iron homeostasis, may promote iron accumulation observed in the brains of KRS patients.

#### 1.4.3. *AP4M1* (OMIM*602296) and *REPS1* (OMIM*614825)

Iron accumulation in the globus pallidus has been recently reported in three patients from a large consanguineous Moroccan family affected by early-onset developmental delay and deterioration of motor function, tetraparesis and intellectual disability. WES analysis of the affected members revealed mutations in a gene denominated *AP4M1* [109]. Mutations in this gene have been previously described in cases with severe congenital microcephaly [110], and in an autosomal-recessive type of tetraplegic cerebral palsy with mental retardation, reduction of cerebral white matter, and atrophy of the cerebellum [111]. The clinical and neuroradiological features of the Moroccan family are not only very different from the previously reported cases, but represent the first time in which iron accumulation is documented in association with *AP4M1* mutations.

*AP4M1* encodes for the mu subunit of the heterotetrameric adaptor protein complex-4 (AP-4), which is ubiquitously expressed in the CNS and involved in vesicle formation, post-Golgi protein trafficking, and sorting processes. AP-4 dysfunction might impair endosomal formation and affect autophagic process, which has already been demonstrated to play a pathogenic role in other NBIA forms related to *ATP13A2* and *WDR45* mutations. 

Mutations in *REPS1* were identified by exome sequencing in two sisters, of unrelated healthy parents, presenting with typical clinical features of NBIA and iron overload in the globus pallidi and peduncles revealed by T2* evidence. MRI also showed progressive cerebellar and cerebral atrophy. *REPS1* codes for a protein involved in endocytosis and vesicle transport [94]. As previously discussed for the *CRAT* gene [92], a defective TfR1 palmitoylation is present in fibroblasts derived from *REPS1* mutant patients, and artesunate treatment is able to increase TfR1 palmitoylation and decrease steady-state ferritin levels, which could be considered as indirect evidence of total iron content reduction [92]. 

### 1.5. Iron Homeostasis in Disease Genes with Unknown Functions 

#### 1.5.1. *DCAF17* (OMIM*612515)

Woodhouse-Sakati syndrome (WSS) (OMIM #241080) is a rare disease transmitted as autososmal recessive trait and characterized by a combination of movement and metabolic impairment. Iron overload is present in the *GP* and *SN* in a fraction of affected patients [112]. The disorder is caused by mutations in *DCAF17* (*C2orf37*) gene, located on chromosome 2q31.1 and encoding for a protein located in the nucleolus. *DCAF17* is the acronym for DDB1 and CUL4-associated factor 17 and the gene belongs to the *DCAF* gene family, which encode receptor proteins for specific ubiquitin ligases involved in DNA damage and cell cycle control [113], but the exact role of the DCAF17 protein remains to be defined.

*Dcaf17* constitutive knockout mice showed male infertility due to abnormal sperm development [114]. An additional mouse model generated by CRISPR/Cas9 approach determining loss of function mutation in exon 2 of *Dcaf17*, showed also female subfertility in addition to male infertility [115]. This phenotype recapitulates hypogonadism and infertility, which are consistent findings in patients carrying *DCAF17* mutations [112] and indicates that this gene plays a role in mammalian gonadal development. It is however, not clear why loss of function mutations of *DCAF17* lead to the complex clinical presentation of affected patients and why knock-out mice only develop infertility and no other signs typical of the human disorder.

#### 1.5.2. *GTPBP2* (OMIM*607434)

Mutations in this gene have been reported for the first time in four subjects presenting with mental retardation, ataxia and dystonic features. Cerebellar atrophy and hypointensity in the *GP* and *SN* suggestive of iron overload were detected by MRI [116]. More recently, biallelic inactivating variants in the same gene have been reported in three unrelated families with a neurological phenotype different from the previously published family and displaying agenesia of the corpus callosum without signs of iron accumulation [117]. The *GTPBP2* is located on human chromosome 6.p21.1 and encodes for a GTP-binding protein, which could play a role as regulator and adaptor of the exosome-mediated mRNA turnover pathway [118]. 

In mice, a homozygous *GTPBP2* mutation interfering with the normal splicing process, determined a neurodegenerative phenotype characterized by apoptosis of neurons accompanied by locomotor dysfunctions, and degeneration of the retinal neurons [119]. These features are evident only in a mouse strain harboring an additional mutation in a gene, expressed in the brain and encoding tRNAArg_UCU_, indicating that GTPBP-2 may play a role in mRNA surveillance and ribosome-associated quality control. A recent study aimed at understanding the role of GTPBP1 and GTPBP2, demonstrates that although GTPBP2’s function involves interaction with aa-tRNA, it deos not stimulate exosomal degradation and have a different function as compared to GTPBP1 [120]. 

## 2. Conclusions

Iron overload in the brain is a distinct finding in rare neurodegenerative disorders classified as NBIA, but is also observed in common neurodegenerative disorders, including Parkinson’s, Alzheimer’s, and ALS. Two NBIA gene products play a direct role in iron homeostasis, but whether other NBIA-related gene products also control iron metabolism in the human brain, and whether high levels of iron contribute to the progression of neurodegeneration is still a matter of investigation. An interesting study based on a systems biology approach aimed at clarifying the pathogenic mechanisms of these disorders and identifying the network connecting known NBIA transcripts with specific partners. This analysis indicated that multiple cell types contribute to the clinically heterogeneous group of NBIA disorders, revealed strong links with iron metabolism and demonstrated the presence of common pathways shared by NBIAs and overlapping neurodegenerative disorders [121].

More recently, interesting unifying mechanisms relating altered iron metabolism to the different NBIA sub-types start to be experimentally proved. The impairment of transferrin receptor palmitoylation and recycling was demonstrated to be common to fibroblasts derived from patients carrying mutations in *PANK2*, *PLA2G6*, *C19orf12*, *FA2H*, and in two recently identified disease genes *CRAT* and *REPS1* [94]. An altered recycling of transferrin receptor was also independently demonstrated in *WDR45* mutant fibroblasts [99]. Additional investigations revealed that iron accumulation might be also related to dysfunctions of neural cells caused by alterations of mitochondrial activities, lipid metabolism, membrane remodeling, and autophagy [122]. Alteration of the structure of mitochondrial cristae is a common feature in NBIA sub-types [60,71] and can cause energy deficiency determined by defective assembly of respiratory chain super complexes [53,64]. 

Mitochondria and their interaction with the endoplasmic reticulum play a crucial role in lipid metabolism and in the formation of autophagosomes. Finally, mitochondria are crucial organelles for the regulation of iron metabolism in other neurodegenerative disorders, such as Parkinson’s and Alzheimer’s, but also Friedreich ataxia and X-linked sideroblastic anaemia and ataxia [72]. 

During the last years the research in this field have massively contributed to the clarification of the pathogenic mechanisms, which are now emerging specially for the most frequent causes of NBIA. Nevertheless, the intricate connection between all the different players acting in NBIA remains to be clarified, paving the way to the identification of tailored approaches to therapy. 

## Figures and Tables

**Figure 1 pharmaceuticals-12-00027-f001:**
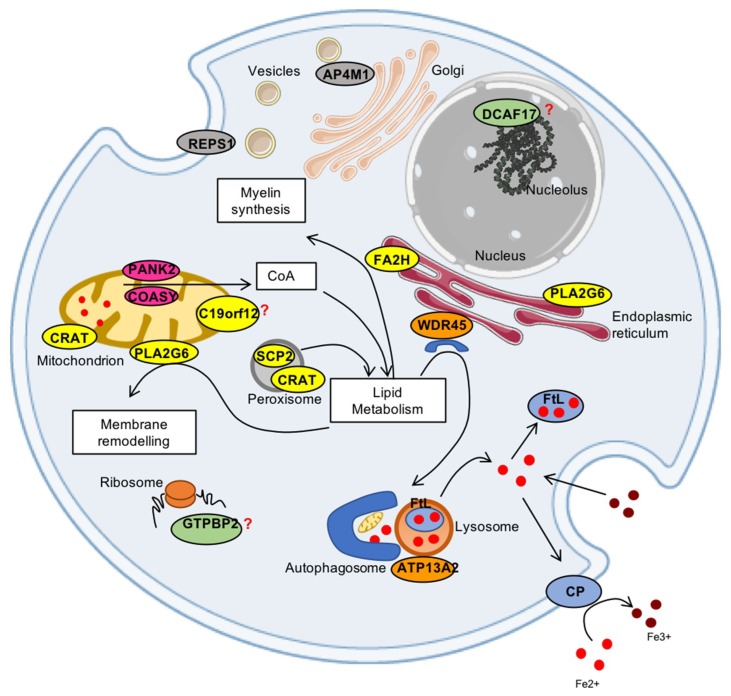
Scheme of the proteins associated to NBIA disorders and their cellular localization. The iron proteins (CP and FtL) are represented in light blue; in pink are the proteins (PANK2 and COASY) are involved in CoA synthesis; in yellow are the proteins related to lipid metabolism (PLA2G6, FA2H, SCP2, CRAT, C19orf12); in orange are the proteins (WDR45, ATP13A2) involved in autophagy; in grey are the proteins (RESP1 and AP4M1) associated to vesicle trafficking; and the proteins (DCAF17 and GTPBP2) in green still have unknown functions.

**Table 1 pharmaceuticals-12-00027-t001:** Molecular and clinical features of NBIA sub-types.

Gene	Disease	Inheritance	Function	Protein localization	Brain Iron and Clinical Features
**Genes related to iron homeostasis**					
*CP*	Acerulo-plasminaemia	AR	Iron oxidation	Plasma membrane	Iron in the basal ganglia, liver, pancreas and myocardium. Movement disorders, dementia, diabetes mellitus, retinal degeneration, dysarthria, ataxia
*FTL1*	Neuro-ferritinopathy (NF)	AD	Cellular iron storage	Cytoplasm	Iron deposition in basal ganglia, cerebellum, motor cortex; mild cerebral and cerebellar atrophy, cavitation of the putamen. Extrapyramidal movement disorders, dystonia, parkinsonisms, dysarthria
**Genes related to Coenzyme A biosynthesis**					
*PANK2*	Pantothenate kinase-associated neurodegeneration (PKAN)	AR	Panthotenate phosphorylation; Coenzyme A synthesis	Mitochondria (inner membrane space)	Iron overload in *GP*; “eye of the tiger sign”. Dystonia, spasticity, cognitive decline, pigmentary retinopathy
*COASY*	COASY protein-associated neurodegeneration (CoPAN)	AR	4’-PP adenyltran-sferase and dephospho-CoA kinase; Coenzyme A synthesis	Mitochondria (matrix), cytosol	Iron overload in *GP*. Oro-mandibular dystonia, dysarthria, spastic-dystonic paraparesis, obsessive-compulsive behavior
**Genes related to lipid metabolism**					
*PLA2G6*	PLA2G6-associated neurodegeneration (PLAN)	AR	Hydrolysis of ester bonds at the sn-2 position of phospho-lipids; Membrane remodeling	Mitochondria, endoplasmic reticulum, cytosol	Iron overload in *GP* in <50% of cases. Infantile neuroaxonal dystrophy, hypotonia, gait disturbance and cerebellar atrophy. Dystonia, spasticity and parkinsonisms in adulthood
*FA2H*	Fatty acid hydroxylase-associated neurodegeneration	AR	Hydroxylation of fatty acids; Ceramide synthesis; Myelin formation	Endoplasmic reticulum	Iron overload in *GP* and *SN*. Profound ataxia, dystonia, dysarthria, spastic quadriplegia, axial hypotonia, optic atrophy
*C19orf12*	Mitochondrial membrane protein-associated neurodegeneration (MPAN)	AR	Unknown; Lipid metabolism? Membrane remodeling?	Mitochondrial membranes, endoplasmic reticulum, MAM	Iron overload in *GP* and *SN*; abundant Lewy bodies. Global developmental delay, dystonia, parkinsonism, psychiatric symptoms, spastic paraparesis
*SCP2*	Leukoencephalopathy with dystonia and motor neuropathy	AR	Thiolase activity; Breakdown of branched chain fatty acids	Peroxisomes	Iron deposition in thalamus; Dystonia and spasmodic torticollis, spinocerebellar ataxia, balance and gait impairment
*CRAT*		AR	Carnitine acetyltrasnferase, -oxidation	Mitochondria	Iron accumulation in *GP* and *SN*. Slowly progressive spinocerebellar degeneration. Cerebellar atrophy and posterior leukodystrophy
**Genes related to autophagy**					
*WDR45*	β-propeller-associated neurodegeneration (BPAN)	X-linked (*de novo* mutations)	Protein-protein interaction; Early autophagosome formation	Endoplasmic reticulum	Iron overload in *GP* and *SN*. Global developmental delay, neurological deterioration, dystonia, parkinsonism, cognitive decline, seizures
*ATP13A2*	Kufor-Rakeb disease (KRS)	AR	Lysosomal cation pump; autophagosome formation	Lysosome, mitochondria	Often no iron overload. Early onset parkinsonism, pyramidal signs, altered eye movements, dementia
*AP4M1*		AR	Vesicle formation	Endosome	Iron in globus pallidus reported in a single family. Early-onset developmental delay and deterioration of motor function, tetraparesis, intellectual disability
*REPS1*		AR	Endocytosis and vesicle transport	Cytoplasm, endosome	Iron accumulation in the globus pallidus and peduncles. Trunk hypotonia, progressive cerebellar ataxia, pyramidal syndrome. Cerebellar and cerebral atrophy.
**Genes with unknown function**					
*DCAF17*	Woodhouse-Sakati syndrome (WSS)	AR	Unknown	Nucleolus	Sometimes iron overload in *GP* and *SN*. Extrapyramidal symptoms, dystonia, cognitive impairment, hypogonadism, alopecia, diabetes mellitus
*GTPBP2*		AR	Unknown; mRNA/ribosome stability?	Cytoplasm	Iron overload in *GP* and *SN*; cerebellar atrophy. Mental retardation, ataxia and dystonia

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
