# Peer review of "Neurodegeneration with Brain Iron Accumulation Disorders: Valuable Models Aimed at Understanding the Pathogenesis of Iron Deposition"

_pharmaceuticals, 2019, doi:10.3390/ph12010027_

Round 1

Reviewer 1 Report

"Neurodegeneration with Brain Iron Accumulation disorders: valuable models to study the involvement of iron in neurodegeneration"  By Levi and Tiranti provides a comprehensive review  on the putative role of iron deposition in the pathogenetic mechanism of most NBIA. 

Apart for a few grammatical errors that should be corrected for clarity - as  examples lines: 13, 196 etc I recommend that the review article be accepted for publication.

Author Response

Reply to reviewer #1:

We thank reviewer #1 for the positive evaluation of our manuscript. 

We reviewed the paper for grammar and syntactical errors.

Reviewer 2 Report

General comments:

This is a potentially exciting and well-organized review but the overall focus is confusing and should be clarified. In the abstract and title, the authors focus seems to be the role of iron in the pathology. However, in the body of the review the authors focus on discussing what can be  “learned from available in vivo and in vitro models about iron metabolism and its connection with the different genetic cause of NBIA”. These seem to be different topics. The authors should either focus on one of these topics or extend specific areas pertinent to these topics. For example:

In respect to the first topic (role of iron) a more substantial discussion in each disease of whether and how iron overload defines or influences the disease initiation or/and progression would be needed. As it stands, the role if iron in the pathology and/or disease progression is not clearly addressed despite an overall detailed description of the disease pathology itself. For the most part, it is only apparent that iron overload can be seen in the various brain regions to various extend, but its role remains largely un-discussed.

In respect to the second focus where the authors try to make connections of iron overload to iron unrelated protein disruptions, a link is made of iron overload and mitochondrial dysfunction, but the underlying mechanism that would substantiate such a link are not well discussed. It would also have been interesting to highlight the “missing  data” or the limitation of the approaches that still contribute to a rudimentary understanding to why many of the NBIAs show iron overload in the first place.

Specific issues:

Line 26-28: Please provide references for regional iron accumulation

Line 76: Please explain the “eye of the tiger sign”

Line 78-81: The authors state that the intracytoplasmic bodies contain ferritin and iron. Do the authors refer here to iron bound in ferritin and/or the labile iron pool?

Line 104-113. It should be explained whether the increase in cellular iron (due to dysfunctional ferritin) prompts the expected down-regulation of proteins involved in cellular iron uptake to counterbalance the increased iron levels. If such a response does indeed occur, why is it not sufficient to limit intracellular iron influx and thus calibrate the system? In addition, please provide some data or references that support the hypothesis that “the unbound metal precipitating on proteins promotes their aggregation”.

Line 117: Please discuss the drastically various outcomes of iron chelation in mice and human patients.

Line 119. The authors state that CP represents the “first disease in which the alteration of an iron related protein has been linked to neurodegeneration”. Consequently, it would make more sense to discuss this disease first.

Line 124: The authors refer to “50 distinct causative mutations”. Are all these distinct mutations in the CP gene itself? Please clarify

Line 160: It is not clear why “simultaneous ablation of CP and hephaestin (HEPH ) is required in mice to recapitulate symptoms consistent with those shown by aceruloplasminemia patients

who have no mutations in HEPH. Please discuss. Again, please provide some hypothesis of why iron chelation seems to fail in humans. (Could be discussed together with NF (line 117)).

Minor corrections:

Line 13: missing word – should read “They are heterogenous in regard…”

Line 15: Missing word- should read “In the region of the basal ganglia ….”

Line 16: I assume the author means “code” when referring to genes that encode specific proteins. The meaning of “codify” is “classify, sort or categorize.

Line 58: change “genetic cause” to “underlying genetic mutations”

Line 85: missing the word ”the"

Author Response

Reply to reviewer #2:

We thank reviewer #2 for the stimulating comments on the manuscript.  In this revised version we addressed all the comments.

Major comment:

We revised the title and the abstract to better focus the aim of the review.

Specific issues:

Line 26-28: Please provide references for regional iron accumulation

The reference was added in the text.

Line 76: Please explain the “eye of the tiger sign”

The explanation has been added in the text:

 “……..a hypointense MRI signal, which is pathognomonic for iron accumulation, surrounding a central region of hyperintensity, which is due to gliosis and/or necrosis, in the anteriomedial globus pallidus on T2* sequence .”

Line 78-81: The authors state that the intracytoplasmic bodies contain ferritin and iron. Do the authors refer here to iron bound in ferritin and/or the labile iron pool?

In the text was added the specificity of form of iron and ferritin.

“…the presence of intranuclear and intracytoplasmic bodies containing ferritin and iron in insoluble form.”

Line 104-113. It should be explained whether the increase in cellular iron (due to dysfunctional ferritin) prompts the expected down-regulation of proteins involved in cellular iron uptake to counterbalance the increased iron levels. If such a response does indeed occur, why is it not sufficient to limit intracellular iron influx and thus calibrate the system? In addition, please provide some data or references that support the hypothesis that “the unbound metal precipitating on proteins promotes their aggregation”.

The following sentence was added in the text:

“These events occur despite that the regulation system of iron entry into the cell seems to work properly [33], in fact the level of the main iron-importer protein, the transferrin receptor, results strongly reduced in different experimental models [32-34]. Thus apparently, iron may be efficiently internalized by transferrin/transferrin-receptor independent pathway in neuronal compartment.”

And the appropriate reference regarding “the unbound metal precipitating on proteins promotes their aggregation”. was added.

Line 117: Please discuss the drastically various outcomes of iron chelation in mice and human patients.

The sentence was changed:

“…..but it was ineffective in reducing the brain iron accumulation and thus in controlling neurodegeneration [12],[26].”

Line 119. The authors state that CP represents the “first disease in which the alteration of an iron related protein has been linked to neurodegeneration”. Consequently, it would make more sense to discuss this disease first.

The text was changed following this suggestion.

Line 124: The authors refer to “50 distinct causative mutations”. Are all these distinct mutations in the CP gene itself? Please clarify

The specification was added together with the indication of the web site where the gene mutations are listed.

Line 160: It is not clear why “simultaneous ablation of CP and hephaestin (HEPH ) is required in mice to recapitulate symptoms consistent with those shown by aceruloplasminemia patients

The following sentence was added to the text:

“…suggesting that hephaestin can compensate the lack of CP in mouse but not in human.”

who have no mutations in HEPH. Please discuss. Again, please provide some hypothesis of why iron chelation seems to fail in humans. (Could be discussed together with NF (line 117)).

The paragraph was changed in :

As already discussed for the aceruloplasminemia, also in this case the outcome of the treatment with iron chelators is not as effective in humans. Iron-chelating based therapies in NF patients had severe detrimental effects on systemic iron level causing iron depletion without improving clinical conditions [29],[43]. This may be due to a lower penetrability of the iron chelator in the basal ganglia of human and/or to a late treatment in humans, which develop symptoms when the brain region has already degenerated.